# Establishment of a Feeding Rate Prediction Model for Combine Harvesters

**Zhenwei Liang** [1,2], **Yongqi Qin** [1] **and Zhan Su** [1,3,*]

1. Key Laboratory of Modern Agricultural Equipment and Technology, Jiangsu University, Ministry of Education, Zhenjiang 212013, China; zhenwei_liang@ujs.edu.cn (Z.L.); 2212116038@stmail.ujs.edu.cn (Y.Q.)
2. Ruiniu Stock Company Limited, Wuxi 214000, China
3. Key Laboratory of Crop Harvesting Equipment Technology of Zhejiang Province, Jinhua Polytechnic, Jinhua 321017, China
* Correspondence: sz_627@126.com

**Abstract:** Feeding rates serve as a vital indicator for adjusting the working parameters of the combine harvester. A non-invasive diagnostic approach to predicting the feed rates of combine harvesters by collecting vibration signals of the inclined conveyor was introduced in this study. To establish a feed rate prediction model, the correlation between feeding rates and vibration signal characteristics was investigated. Vibration signal characteristics in both the time domain and frequency domain were also analyzed in detail. The RMS (root mean square) value and the total RMS value of the one-third octave extracted from the vibration signal were utilized to establish a feed rate prediction model, and field tests were conducted to verify the model performance. The experimental results indicated that the relative errors of the established model range from 3.1% to 4.9% when harvesting rice. With the developed feed rate prediction system, the control system of the combine harvester can acquire feed rate information in real time, and the working parameters can be adjusted in advance, thereby, it can be expected to greatly enhance the working performance of the combine harvesters.

**Keywords:** combine harvester; vibration characteristics; feeding rate; prediction model; field test

## 1. Introduction

Combine harvesters, renowned as one of the most intricate agricultural machinery types, seamlessly integrate a multitude of functions encompassing cutting, conveying, threshing, cleaning, and collecting [1,2]. Their pivotal role in crop production is indisputable, significantly elevating agricultural productivity. Research indicates that feed rates exert a profound influence on harvesting performance, underscoring the importance of the timely adjustment of pertinent parameters to optimize efficiency [3,4], and feeding rates also serve as one of the important reference parameters for designing crucial working components of combine harvesters, including threshing rotors, cleaning devices, and grain conveying systems [5]. Nonetheless, variations in crop height, density, and forward speed can induce fluctuations in feeding rates to some extent, consequently affecting the vibration characteristics of these components [6]. Harnessing cutting-edge information technologies such as sensors and real-time data analysis facilitates the precise monitoring of fluctuations in feed rates, thereby enabling swift adjustments to the operational parameters of the threshing device and forward speed. This proactive approach not only amplifies operational efficiency but also effectively mitigates grain losses attributable to feed rate variations. Current research on feeding rate monitoring in combine harvesters often relies on monitoring the torque of the header feeder auger, inclined conveyor, threshing rotors, or grain flow to characterize the overall feeding rate [7–11]. Although these methods often exhibit significant lag behind, as the cut crop enters the inclined conveyor before it enters into the threshing rotors, the feed rate information predicted from the inclined conveyor can be collected by the control system in advance, and the information lag behind

can be overcome. Vibration, as a manifestation of mechanical motion, encapsulates vital information regarding the operational state of equipment, making it a valuable diagnostic tool in mechanical engineering. Vibration analysis, a widely adopted diagnostic method, unveils both the operational status and potential structural issues of mechanical equipment [12–14]. The external inputs excite the vibration system, causing vibration responses to manifest at measurement points through various transmission paths. These responses essentially constitute the aggregate of modal responses generated at measurement points under the influence of working loads [15]. When a structural system experiences external excitation, it exhibits natural vibrations at specific frequencies [16,17]. These vibration signals harbor distinct characteristics reflective of the equipment's operational status. By collecting and analyzing vibration signals, it becomes feasible to effectively predict and monitor the equipment's condition and feeding rates.

Currently, research progress has been made in vibration research, and the performance of the combine harvester has improved significantly. Most of the relevant efforts have focused on establishing vibration models [18–21] and utilizing finite element analysis to analyze working components such as headers, cleaning sieves, and threshing rotors [22–24]. Simultaneously, scholars have proposed that differences in feeding rates can affect the intensity of vibrations in the working components of combine harvesters. Ebrahimi et al. identified and alleviated vibration issues in the combine harvester cutting platform through experimental assessment and finite element model updates, further validating the accuracy of frequency domain decomposition technique in vibration characteristics research and reducing vibration levels through structural modifications [25]. Gao et al. conducted vibration tests under varying feeding rates and found differing vibration intensity patterns with increasing feeding rates, particularly observing increased loads on the drive shaft and a positive correlation between total vibrations and feeding rates at certain points [26]. Ding et al. investigated the effects of feeding rate disturbances on combine harvester vibration characteristics and found that vibrations in the cutter and conveyor increased gradually with higher feeding rates during field trials [27]. Yao et al. studied vibration characteristics in corn harvesters, highlighting the influence of factors such as operating conditions and mass changes on vibration behavior [28]. These studies not only deepen the understanding of the vibration characteristics of combine harvesters but also provide important theoretical and practical foundations for optimizing the design and enhancing the reliability of agricultural machinery. However, they primarily focus on investigating the vibration characteristics of combine harvesters and their correlation with factors such as feed rate or operating conditions, without further utilizing vibration characteristics to reflect the working status of combine harvesters. To date, there has been no research capable of analyzing corresponding fluctuations in feed rates by analyzing the vibration signal characteristics of working components in combine harvesters. This work proposes a novel application of vibration signal analysis for the real-time monitoring of feed rates.

The aim of this research is to develop a non-invasive diagnostic approach, which refers to a technique or approach that does not require direct contact or alteration of the surface or interior of an object, to monitor the feed rates of combine harvesters by collecting vibration signals of the inclined conveyor and investigate the correlation between feeding rate fluctuations and the characteristics of the vibration signals. Utilizing the selected vibration signal characteristics, the feed rate monitoring model was established and verified by a field experiment. With the developed feed rate monitoring system, the control system of the combine harvester can acquire feed rate information in time as the cut crop passes through the inclined conveyor before it enters the threshing device, and the working performance can be improved greatly as the working parameters of combine harvesters can be adjusted in advance.

## 2. Materials and Methods

### 2.1. Sensor Installation Position

This study focuses on the impact of feeding rate on the vibration characteristics of the inclined conveyor in a rice combine harvester (type, 4LZ-6.0EK, World Group, Zhenjiang, China). Acceleration sensors were selected as the vibration signal measurement device, and the DH5902 Dynamic signal acquisition instrument was used to collect and analyze the vibration signal. The sensor installation positions in the inclined conveyor are illustrated in Figure 1, with Sensor 1 located on the left side of the bottom of the inclined conveyor, closer to the header, designated as measurement point 1. Sensor 2 is positioned on the right side, closer to the threshing device, designated as measurement point 2. This layout effectively captures the vibration signals of crop flow within the inclined conveyor, aiding in a more comprehensive understanding of crop movement within the inclined conveyor and providing accurate input data for the control system. Although these two sensors are installed in close proximity to each other, considering the working environment of the inclined conveyor and the constraints of the machine's structure, this is the best installation solution we can adopt. During field harvesting operations, various working components of the combine harvester typically operate at rated speeds, especially while maintaining the feed rate. The vibration frequencies generated by the header and threshing device are relatively stable. Based on this, reasonable data processing and vibration feature extraction can be conducted to extract stable characteristic parameters from the vibration signals. Table 1 lists the main parameters of the accelerometer and the dynamic signal analyzer. The DH5902 dynamic signal analyzer used in this study not only reliably captures and stores vibration signals but also provides extensive signal data processing capabilities.

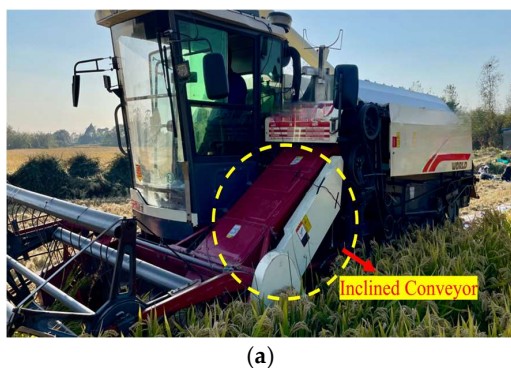 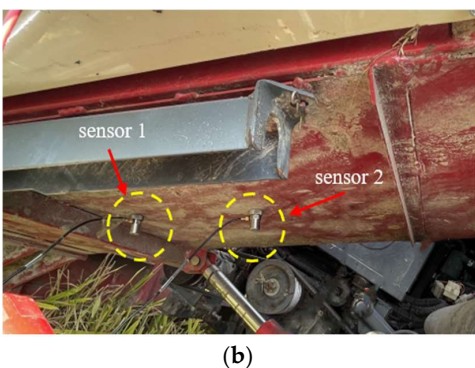

(**a**)          (**b**)

**Figure 1.** Installation location of vibration accelerator sensors. (**a**) The position of the inclined conveyor in the combine harvester. (**b**) The position of the sensors on the inclined conveyor.

**Table 1.** Main performance parameters of sensors and the signal-acquiring device.

| Equipment Name | Performance Index | Parameter Values | Unit |
|---|---|---|---|
| Universal piezoelectric accelerometer | Sensitivity | 5 | $mV \cdot (m \cdot s^{-2})^{-1}$ |
| | Range | 1000 | $m \cdot s^{-2}$ |
| | Frequency | 0.5~7000 ($\pm 10\%$) | Hz |
| DH5902 Dynamic signal acquisition instrument | Number of channels | 32 | |
| | Full scale value | $\pm 20 \sim \pm 20,000$ | mV |
| | Distortion | $\leq 0.5$ | % |
| | Maximum sampling Frequency | 100 | KHz |

Although installing multiple sensors on the inclined conveyor can provide more comprehensive data and enable a more detailed analysis of its vibration characteristics, this poses challenges in data processing. The large volume of data requires more complex data processing and analysis techniques to handle, and the complexity of the feed rate prediction

model will also increase exponentially. Furthermore, field validation experiments have shown that the vibration signals collected by two sensors can accurately reflect the feed rate of crops in the inclined conveyor. Therefore, after weighing the pros and cons in practical applications, it was determined to use only two sensors to meet the research requirements.

To avoid aliasing, the sampling frequency should be at least twice the highest frequency component in the signal, while the analyzed frequency range should not exceed half of the sampling frequency according to the Nyquist theorem. It is indicated that the vibration signals collected on the combine harvester are mainly composed of low-frequency components [29]. During the experiment, the sampling method of the test system was set to continuous sampling, with a sampling frequency of 2.56 kHz. The analysis frequency is equal to the sampling frequency divided by 2.56, resulting in 1000 Hz. Additionally, in the frequency-domain analysis, the number of analysis points is 4096, the number of spectral lines is 1600, and the averaging time of the average spectrum is 10 times. Given the known sampling frequency and number of spectral lines, the length of the time-domain data for each FFT transformation, denoted as $T_0$, can be calculated using the following formula:

$$T_0 = \frac{1}{Frequency\ Resolution} = \frac{Number\ of\ Spectral\ Lines}{Analysis\ Frequency} \tag{1}$$

It can be concluded that $T_0$ is equal to 1.6 s, which means the time-domain data length for each FFT transformation is 1.6 s. Therefore, the minimum length for an average of 10 transformations is 16 s.

### 2.2. Vibration Signal Analysis Methods

To obtain the characteristic information required for feed rate prediction, necessary signal processing techniques such as time-domain analysis, frequency-domain analysis, and wavelet analysis are employed [30–32]. Time-domain analysis elucidates how signal parameters evolve over time, offering an intuitive depiction of signal amplitude. This can be leveraged in vibration signal analysis by computing metrics such as the mean value, RMS (root mean square) value, peak value, and peak-to-peak value. Notably, the time-domain characteristics of non-stationary stochastic signals with different periodicities exhibit discernible disparities [33,34]. Suppose there exists a signal denoted as *X*, with a length of *N* and the value of the *n*th sample point represented as *x(n)*. Among these, the mean value ($\mu$), also known as the mathematical expectation, reflects the signal average amplitude in the time domain, expressed as:

$$\mu = E(X) = \frac{1}{N} \sum_{n=1}^{N} x(n) \tag{2}$$

The mean square root value (*RMS*), known as the average power, is expressed as:

$$RMS(X) = \sqrt{\frac{1}{N} \sum_{n=1}^{N} x^2(n)} \tag{3}$$

The mean square root value represents the signal strength. Its positive square root, known as the effective value, is another representation of the signal's average energy. The advantage of using the effective value to characterize the magnitude of vibration lies in considering both the time course of vibration and the magnitude of vibration energy.

The peak-to-peak value (*PP*), defined as the difference between the maximum and minimum values of the signal, describes the size of the variation range of signal values. It is expressed as:

$$PP(X) = \max(X) - \min(X) \tag{4}$$

The vibration signal induced by feeding rates is an external disturbance that needs to be distinguished from the excitation signal of the combine harvester. Vibration signal

correlation analysis includes auto-correlation analysis and cross-correlation analysis, both of which are time-domain analysis methods. The auto-correlation function reflects the similarity in signal values at different instants and represents the self-correlation of the signal. It provides an average measure of the numerical dependence obtained by comparing two observations of the same signal, revealing useful information in noisy signals, and identifying repeated information or fundamental frequencies. This method can detect periodic vibrations and diagnose the causes of external disturbances [35].

The mathematical definition of the auto-correlation function is as follows:

$$R_{XX}(\tau) = \lim_{T \to \infty} \frac{1}{T} \int_0^T X(T) X(t + \tau) \, \mathrm{d}t \tag{5}$$

The auto-correlation function is the average of the product of the signal $X(t)$ and its time-shifted version $X(t + \tau)$, where $\tau$ represents the time-shift variable.

The cross-correlation function describes the general dependency between two sets of random signals, illustrating the waveform similarity between the signal $X(t)$ and $Y(t)$ after a time shift of $\tau$, denoted as $Y(t + \tau)$. The computational formula is given by:

$$R_{XY}(\tau) = \lim_{T \to \infty} \frac{1}{T} \int_0^T X(T) Y(t + \tau) \, \mathrm{d}t \tag{6}$$

Time-domain signals offer significant advantages in temporal resolution, allowing for the precise depiction of signal variations over time. However, their limitation lies in zero frequency resolution, meaning they cannot provide detailed information about which frequency components the signal contains and how these components affect the signal. To address this limitation, frequency-domain analysis has become an important signal processing technique, revealing the frequency characteristics of a signal by decomposing it into multiple sinusoidal waves [36]. The Fourier transform is the core tool of frequency-domain analysis, where clarity on the frequency components of a signal and their contributions to the overall signal can be gained through Fourier transformation. By combining time-domain and frequency-domain analyses, a more comprehensive understanding of the dynamic changes in signals can be achieved, providing a more scientific basis for machinery condition monitoring and fault diagnosis. The Discrete Fourier Transform (DFT) formula exists in various forms; one common representation is the exponential form, which is given as [37–40]:

$$F(k) = \sum_{n=0}^{N-1} x(n) \, e^{\frac{-j 2 \pi n k}{N}} \tag{7}$$

where:

$F(k)$ represents the kth frequency component in the frequency domain;

$x(n)$ represents the discrete signal in the time domain;

$N$ is the total number of samples in the signal;

$j$ is the imaginary unit;

$k$ ranges from 0 to $N - 1$.

The fast Fourier transform (FFT), commonly used in modern computing, is a fast algorithm for computing the Discrete Fourier Transform (DFT).

### 2.3. Field Test Arrangement and Model Establishment

Feeding rate refers to the mass of crops processed continuously by the combine harvester per unit time. Its influencing factors mainly include the forward speed, the cutting width, and the crop yield, which can be expressed by the following equation:

$$Q = L \cdot v \cdot A(1 + \theta) \tag{8}$$

where $Q$ is the feeding rate of the combine harvester, kg·s$^{-1}$; $L$ is the cutting width, m; $v$ is the forward speed, m·s$^{-1}$; $A$ is the crop yield per unit area, kg·m$^{-2}$, $\theta$ is ratio of material-other-than-grain (MOG) to grain.

To explore the effect of feeding rates on the vibration characteristics of the inclined conveyor during field harvesting, in this study, we selected a field with more even crop density as the test area, and a test was conducted in Dantu District, Zhenjiang City, Jiangsu Province, China. The basic characteristics of the experimental rice are shown in Table 2. Maintaining the maximum cutting width for harvesting and keeping the header height unchanged, vibration signals were collected at the inclined conveyor by varying the forward speed to change the feed rate. The test arrangements are shown in Table 3.

**Table 2.** Basic characteristics of the rice used in the field experiments.

| Items | Yangnong No. 1 |
|---|---|
| Grain moisture content/% | 22.5 |
| Stem moisture content/% | 76.5 |
| Crop natural height/cm | 85.2 |
| MOG/grain ratio | 2.0 |
| Thousand grain mass/g | 30.0 |
| Yield per unit area/kg·m$^{-2}$ | 0.74 |

**Table 3.** Vibration signal acquisition arrangement.

| Experiment Arrangement | Test Distance (m) | Forward Speed (m·s$^{-1}$) | Feeding Rate (kg·s$^{-1}$) |
|---|---|---|---|
| Group 0 | 0 | 0 | 0 |
| Group 1 | 25 | 0.3 | 1.47 |
| Group 2 | 25 | 0.6 | 2.94 |
| Group 3 | 25 | 0.9 | 4.41 |

In Experiment Group 0, the feeding rate of the combine harvester was set to zero, with the engine and working components operating simultaneously, and vibration signals at the inclined conveyor under this no-load condition were collected. Experiment Group 3 maintained a forward velocity of 0.9 m·s$^{-1}$, where the feeding rate reached its maximum at 4.41 kg·s$^{-1}$. Experiment Groups 2 and 1 had progressively reduced forward velocities of 0.6 m·s$^{-1}$ and 0.3 m·s$^{-1}$, respectively, resulting in corresponding feeding rates of 2.94 kg·s$^{-1}$ and 1.47 kg·s$^{-1}$. Each test was conducted three times.

## 2.4. Feeding Rate Prediction Model Establishment and Field Experiment Verification

After time-domain and frequency-domain analyses of the collected vibration signals, the proper signal characteristics and multiple linear regression method were used to establish a model to predict the feeding rate of the combine harvester. To simplify calculations, the "Linear Regression" class of the scikit learn library in Python was used to create the model. Then, the model validation was carried out to compare the predicted values from the model with the theoretical values. Lastly, the field validation experiment was conducted under conditions consistent with the previous experiments. Other experimental conditions, such as the type and installation position of vibration sensors, were kept largely consistent, except for the forward velocity to minimize the influence of external factors. In the field validation experiment, five validation groups were conducted, and the forward velocity of the combine harvester was controlled between 0.2 m·s$^{-1}$ and 1.0 m·s$^{-1}$. The harvesting distance was kept consistent at 25 m. After completing the experiments, the data were imported into the analysis software, and time-domain and frequency-domain analyses were conducted with the same setting parameters. The required feature values were extracted and plugged into the feeding rate prediction model to achieve feeding rate prediction. Finally, the predicted values were compared with the theoretically estimated feeding rates to analyze the errors.

## 3. Results and Analysis

### 3.1. Time-Domain Analysis of Each Experimental Group

Taking Experiment Groups 0 to 4 as examples, the vibration signals collected by the vibration acceleration sensors under different working conditions are illustrated within the time domain, as shown in Figure 2. The horizontal axis represents the sampling time, while the vertical axis represents the acceleration. An observation of the signal variations from Experiment Groups 0 to 4 clearly indicates a gradual increase in vibration signals with the increase in feeding rate. To further analyze the trend in vibration signals collected from each experimental group, certain features were extracted from the time-domain signals, including the mean, peak-to-peak value, standard deviation, and RMS. These features provide more detailed insights into the temporal variations in the vibration signals. The time-domain feature parameters of the vibration signals collected by Sensor 1 and Sensor 2 are listed in Table 4. From Table 4, it is evident that in each experimental group, the peak-to-peak value, standard deviation, and RMS of Sensor 2 are consistently significantly greater than Sensor 1. This indicates that the intensity of the vibration signal measured at Sensor 2 is always greater than that at Sensor 1 under the same conditions. This discrepancy is attributed to Sensor 2 being closer to the active axis of the conveyor, resulting in larger vibration displacements and consequently higher vibration signal intensities. Such differences need to be taken into account in data analysis to ensure that the specific sensor installation positions are considered when interpreting changes.

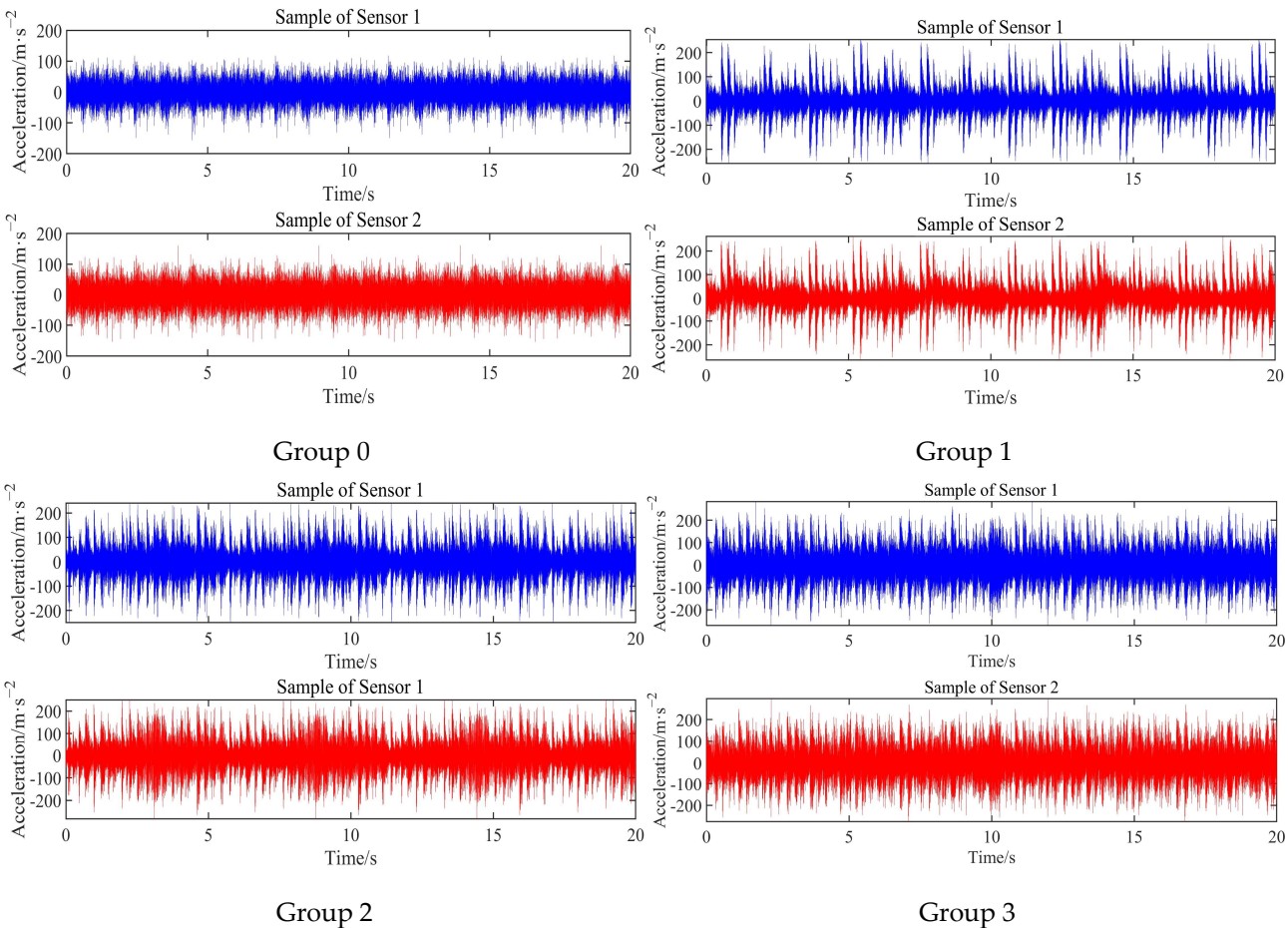

**Figure 2.** Time-domain signals of each experimental group.

It is also observed in Table 4 that the peak-to-peak values gradually increase from Experiment Group 0 to Experiment Group 4. At Sensor 1, the peak-to-peak value rises from 303.92 m·s$^{-2}$ to 554.01 m·s$^{-2}$, while at Sensor 2, it increases from 317.05 m·s$^{-2}$ to 573.32 m·s$^{-2}$. Since the peak-to-peak value represents the amplitude range of the waveform,

this indicates that the amplitude range of the vibration signal increases with the feeding rate. However, this increase is not linear, which is particularly evident when the feeding rate increases from 1.47 kg·s$^{-1}$ to 2.94 kg·s$^{-1}$, where the peak-to-peak values remain nearly the same. Furthermore, even as the feeding rate escalates to 4.41 kg·s$^{-1}$, there is only a slight increase in the peak-to-peak values. This suggests that the peak-to-peak value may not provide favorable insight into changes in the feeding rate.

**Table 4.** Time-domain characteristic parameters of each experimental group (m·s$^{-2}$).

| Experiment Scheme | Measurement Point | Mean | Peak-to-Peak | Standard Deviation | Root Mean Square |
|---|---|---|---|---|---|
| Group 0 | 1 | −1.39 | 303.92 | 31.88 | 31.91 |
| | 2 | −1.14 | 317.05 | 37.29 | 37.31 |
| Group 1 | 1 | −1.19 | 513.69 | 49.49 | 49.50 |
| | 2 | 4.60 | 525.79 | 50.28 | 50.31 |
| Group 2 | 1 | −1.16 | 504.02 | 58.60 | 58.61 |
| | 2 | −2.47 | 532.21 | 62.82 | 62.84 |
| Group 3 | 1 | −1.16 | 554.01 | 64.03 | 64.04 |
| | 2 | −1.36 | 573.32 | 68.07 | 68.09 |

The peak-to-peak value only provides the extreme amplitude range of the vibration waveform and is insensitive to changes in the overall shape of the waveform. While RMS values consider the direction of the signal, they can more comprehensively reflect the magnitude of the vibration waveform. For complex waveforms, RMS values more accurately represent the effective amplitude of the signal and are more sensitive to changes in the waveform. An increase in an RMS value indicates a gradual increase in the vibration amplitude. This increase may result from increased force in the mechanical system, changes in excitation sources, or other external stimuli. Moreover, for vibration signals, the RMS value can be considered as the average vibration energy of the signal, reflecting the distribution of signal energy. Therefore, a gradual increase in the RMS value indicates a gradual increase in vibration signal energy.

In Table 4, it is evident that as the feeding rate gradually increases, the RMS values of the signals also increase gradually. At Sensor 1, the RMS value increases from 31.91 m·s$^{-2}$ to 64.04 m·s$^{-2}$, while at Sensor 2, it increases from 37.31 m·s$^{-2}$ to 68.09 m·s$^{-2}$. This experiment demonstrates that the vibration energy of the signals collected by both sensors gradually increases with the increase in feeding rate.

### 3.2. Frequency-Domain Analysis of Each Experimental Group

Time-domain analysis is suitable for observing the changes in vibration signals over time, while frequency-domain analysis provides important information about the frequency characteristics of the signals. To analyze the different frequency components and their relative strengths contained in the signal, the fast Fourier transform (FFT) is used to convert the time-domain signal into the frequency-domain signal, generating a spectrum plot that displays the amplitudes of various frequency components in the signal. As shown in Figure 3, the spectrum plot typically has frequency on the horizontal axis and acceleration magnitude on the vertical axis. From the spectrum plots of each experimental group, the first four peak amplitudes and their corresponding vibration frequencies are extracted and presented in Table 5.

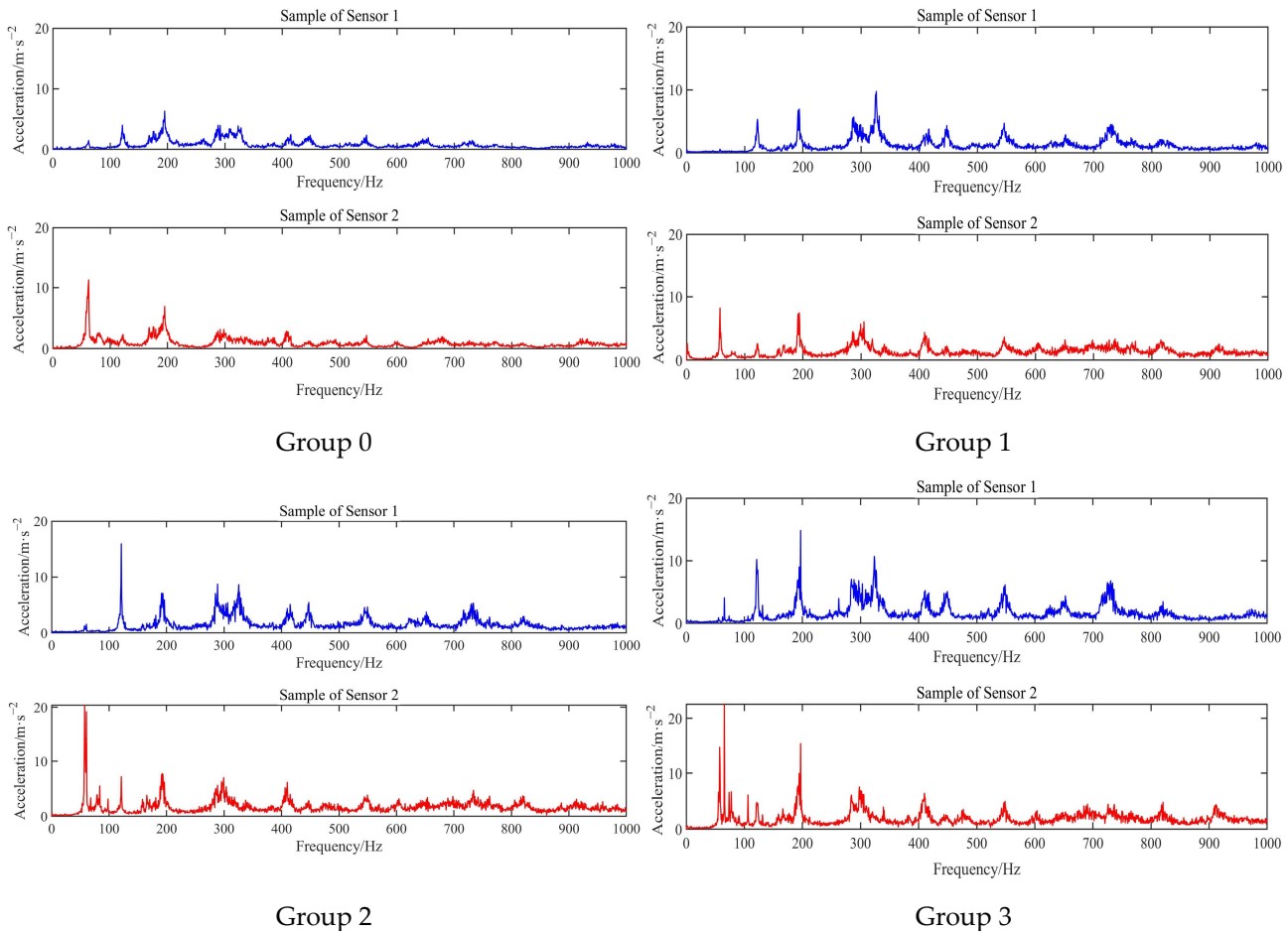

**Figure 3.** Spectral charts of each experimental group.

**Table 5.** Amplitude frequency and peak value of each experimental group.

| Experiment Scheme | Measurement Point | Peak 1 | | Peak 2 | | Peak 3 | | Peak 4 | |
|---|---|---|---|---|---|---|---|---|---|
| | | Frequency (Hz) | Amplitude (m·s$^{-2}$) | Frequency (Hz) | Amplitude (m·s$^{-2}$) | Frequency (Hz) | Amplitude (m·s$^{-2}$) | Frequency (Hz) | Amplitude (m·s$^{-2}$) |
| Group 0 | 1 | 195.00 | 6.34 | 193.75 | 4.91 | 121.25 | 4.02 | 288.125 | 4.01 |
| | 2 | 62.50 | 11.25 | 60.00 | 8.47 | 195.00 | 6.92 | 58.125 | 6.16 |
| Group 1 | 1 | 326.25 | 9.77 | 325.00 | 9.30 | 193.75 | 6.98 | 191.875 | 6.78 |
| | 2 | 57.50 | 8.20 | 193.75 | 7.47 | 191.875 | 7.30 | 305.00 | 6.06 |
| Group 2 | 1 | 121.25 | 15.97 | 288.75 | 8.76 | 325.625 | 8.67 | 326.875 | 7.31 |
| | 2 | 57.50 | 20.41 | 60.625 | 19.21 | 192.5 | 7.80 | 193.75 | 7.63 |
| Group 3 | 1 | 196.875 | 14.85 | 323.75 | 10.71 | 121.25 | 10.23 | 194.375 | 9.03 |
| | 2 | 65.625 | 22.56 | 196.875 | 15.49 | 57.50 | 14.65 | 194.375 | 10.00 |

To provide a clearer and more insightful analysis of the frequency characteristics of vibration signals under varying feeding rate conditions, a one-third octave band analysis of the frequency range from 2 to 1000 Hz was conducted on each spectrum plot to obtain one-third octave bands. The specific data for each experimental group were exported from the dynamic signal testing and analysis software and plotted as histograms, as shown in Figure 4. It is worth noting that the plots represent RMS values, with frequency on the horizontal axis and RMS on the vertical axis. Table 6 illustrates the one-third octave bands of the spectrum for Sensor 1 in Experiment Group 0 within the frequency range of 2 to 1000 Hz, displaying the center frequencies and amplitudes of each sub-band. Next, the data were processed to calculate the average, maximum, and total RMS values for both sensors in each experimental group, as detailed in Table 7. The total RMS reflects the overall vibration intensity across the entire frequency range, while the RMS of each sub-band

provides more detailed frequency-domain information, reflecting the energy distribution across different sub-bands.

$$RMS_{ALL} = \sqrt{\sum_{j=1}^{M} (RMS_j)^2} \tag{9}$$

where *M* is the number of sub-bands in the one-third octave band analysis, dividing the frequency range from 2 Hz to 1000 Hz into a total of 28 sub-bands, and $RMS_j$ represents the RMS value of the *j*-th sub-band.

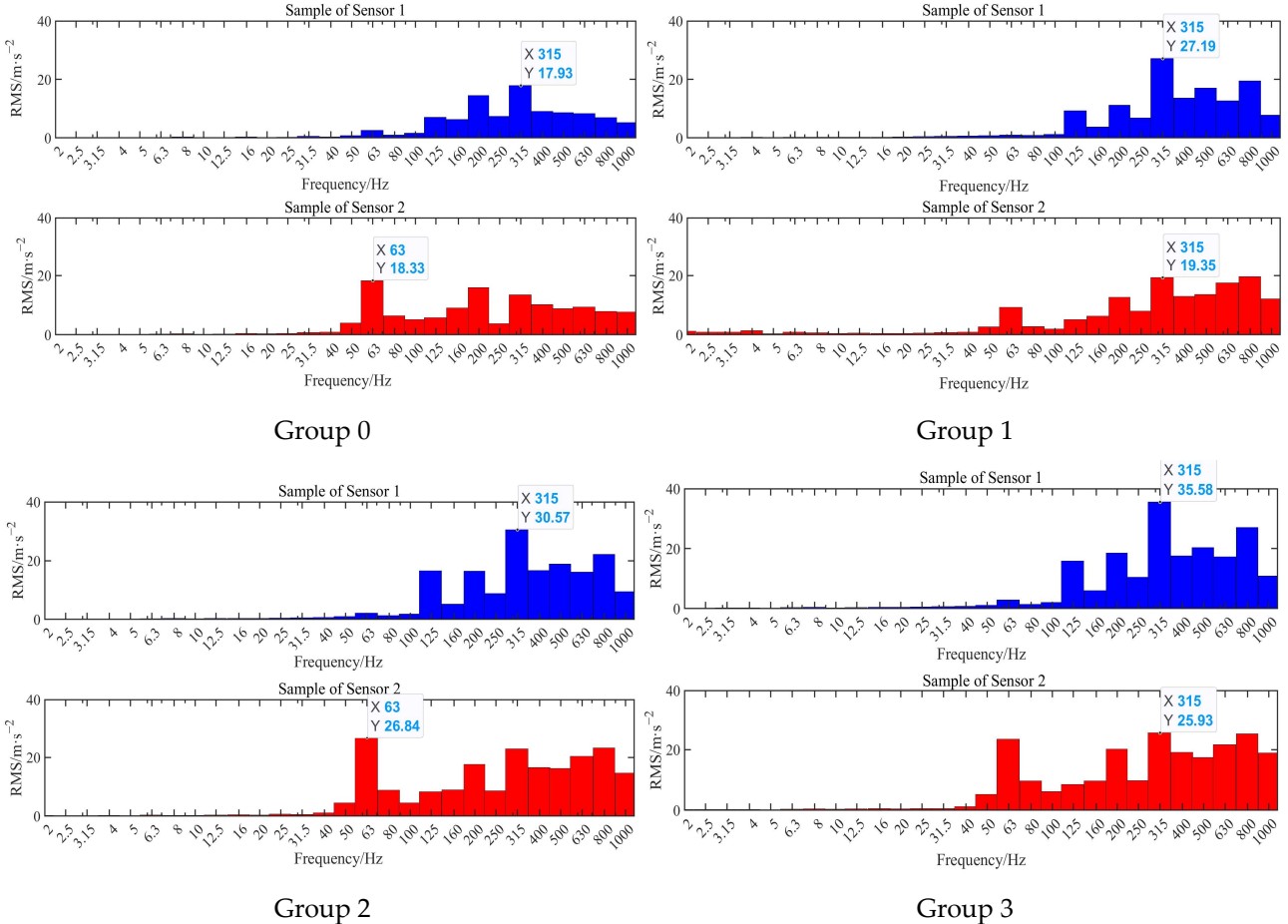

**Figure 4.** One-third octave frequency plots of each experimental group.

**Table 6.** One-third octave bands of measurement point 1 in Experiment Group 0.

| Center Frequency (Hz) | RMS (m·s$^{-2}$) | Center Frequency (Hz) | RMS (m·s$^{-2}$) | Center Frequency (Hz) | RMS (m·s$^{-2}$) | Center Frequency (Hz) | RMS (m·s$^{-2}$) |
|---|---|---|---|---|---|---|---|
| 2 | 0.05 | 10 | 0.08 | 50 | 0.84 | 250 | 7.43 |
| 2.5 | 0.12 | 12.5 | 0.19 | 63 | 2.56 | 315 | 17.93 |
| 3.15 | 0.08 | 16 | 0.41 | 80 | 1.03 | 400 | 9.14 |
| 4 | 0.12 | 20 | 0.17 | 100 | 1.69 | 500 | 8.64 |
| 5 | 0.08 | 25 | 0.31 | 125 | 7.1 | 630 | 8.36 |
| 6.3 | 0.2 | 31.5 | 0.55 | 160 | 6.3 | 800 | 6.97 |
| 8 | 0.35 | 40 | 0.39 | 200 | 14.51 | 1000 | 5.27 |

**Table 7.** Octave band analysis statistical values for each experimental group.

| Experiment Scheme | Measurement Point No. | Mean | Maximum | RMS$_{All}$ |
|---|---|---|---|---|
| Group 0 | 1 | 3.60 | 17.93 | 31.52 |
|  | 2 | 4.66 | 18.33 | 37.19 |
| Group 1 | 1 | 4.89 | 27.19 | 45.81 |
|  | 2 | 5.54 | 19.72 | 44.48 |
| Group 2 | 1 | 6.13 | 30.57 | 55.41 |
|  | 2 | 7.42 | 26.84 | 60.15 |
| Group 3 | 1 | 6.88 | 35.58 | 62.46 |
|  | 2 | 8.14 | 25.93 | 64.94 |

In Figure 4, the height of each bar represents the RMS value of each one-third octave band, which can be used to indicate the energy intensity of all frequency components within that band. Observing Figure 4, from Experiment Group 0 to Experiment Group 3, it is visually evident that the area under the curve for both Measurement Point 1 and Measurement Point 2 gradually increases. Table 7 indicates that the average energy in the frequency range of 2 Hz to 1000 Hz at both measurement points gradually increases as the feeding rate increases. Moreover, the average energy at Measurement Point 2 is consistently greater than that at Measurement Point 1. On the other hand, considering the analysis from the perspective of the maximum RMS value, in octave band analyses, if the RMS value of a particular band remains the maximum among all bands, it indicates that this band is the most concentrated energy part of the signal, implying that the signal has the strongest vibration intensity or the highest energy level in this frequency range. In summary, the significance of the RMS value of a particular band in octave band analyses indicates the importance of that band in the entire signal.

In Figure 4, it is evident that the height of the octave band centered at 315 Hz for Measurement Point 1 remains consistently maximal across all experimental groups, while for Measurement Point 2, the heights of the octave bands centered at 63 Hz and 315 Hz are notably prominent. This indicates that the octave band centered at 315 Hz dominated, consistently providing higher vibration intensity within its frequency components. Focusing on the amplitude of the octave band at 315 Hz, from Experiment Group 0 to Experiment Group 4, at Measurement Point 1, the amplitude gradually increased from 17.93 m·s$^{-2}$ to 35.58 m·s$^{-2}$, and at Measurement Point 2, it increased from 13.49 m·s$^{-2}$ to 25.93 m·s$^{-2}$. This implies that the vibration intensity of the dominant octave band within the octave band analysis increases with the increase in feeding rate.

Finally, the changes in the overall RMS values were analyzed, which calculate the square root of the sum of squares of the amplitudes of all octave bands, reflecting the overall vibration intensity across the entire frequency range. The increase in the overall RMS values indicates an increase in the overall vibration energy. Looking at the changes in the overall RMS values in the last column of Table 7, at Measurement Point 1, it gradually increased from 31.52 m·s$^{-2}$ to 62.46 m·s$^{-2}$, while at Measurement Point 2, it gradually increased from 37.19 m·s$^{-2}$ to 64.94 m·s$^{-2}$. This again confirms that an increase in the feeding rate leads to an increase in the vibration intensity, consistent with the time-domain analysis results, and the magnitude of the increase trend decreases gradually. Additionally, it is worth noting that the average vibration intensity at Measurement Point 1 is lower than at Measurement Point 2 in Group 1, but the overall vibration intensity is slightly higher. This is because the RMS value is more sensitive to extreme values in the data; the presence of greater variability or extreme values in the data at Measurement Point 1 leads to a higher RMS value, which may not significantly affect the average value.

### 3.3. Feeding Rate Prediction Model Establishment

After time-domain and frequency-domain analyses of the collected signals, the time-domain RMS value and the total RMS value of the one-third octave are used to predict the

feeding rate. We set the RMS of Measurement Point 1 in the time domain as $Y_1$, the RMS in the time domain of Measurement Point 2 as $Y_2$, the total RMS of the octave of Measurement Point 1 as $Y_3$, the total RMS of the octave of Measurement Point 2 as $Y_4$, and the feeding rate as $W$. The calculated relationships are shown in Table 8.

**Table 8.** Mapping of feeding rate to feature values.

| Feature Values | | Feeding Rate (kg·s$^{-1}$) (W) | | | |
| --- | --- | --- | --- | --- | --- |
| | | 0 | 1.47 | 2.94 | 4.41 |
| Time-domain RMS | Measure Point 1 ($Y_1$) | 31.91 | 49.50 | 58.61 | 64.04 |
| | Measure Point 2 ($Y_2$) | 37.31 | 50.31 | 62.84 | 68.09 |
| Total RMS of octave band | Measure Point 1 ($Y_3$) | 31.52 | 45.81 | 55.41 | 62.46 |
| | Measure Point 2 ($Y_4$) | 37.19 | 44.48 | 60.15 | 64.94 |

The multiple linear regression method was used to simultaneously consider the values of $Y_1$, $Y_2$, $Y_3$, and $Y_4$ to predict the feeding rate $W$. Firstly, a predictive model equation was established and constructed, as shown in the following equation, where $W$ is the dependent variable and $Y_1$, $Y_2$, $Y_3$, and $Y_4$ are independent variables:

$$W = \beta_0 + \beta_1 Y_1 + \beta_2 Y_2 + \beta_3 Y_3 + \beta_4 Y_4 \tag{10}$$

where $\beta_0$ is the intercept and $\beta_1$, $\beta_2$, $\beta_3$, and $\beta_4$ are the coefficients.

To simplify calculations, the "Linear Regression" class of the scikit learn library in Python was used to create the model. The calculation results are shown in Table 9. After substituting into Equation (10), Equation (11) is obtained.

$$W = -4.3809 - 0.1876 Y_1 - 0.0872 Y_2 + 0.3987 Y_3 + 0.0283 Y_4 \tag{11}$$

**Table 9.** Specific values of the intercept and coefficients.

| Category | $\beta_0$ | $\beta_1$ | $\beta_2$ | $\beta_3$ | $\beta_4$ |
| --- | --- | --- | --- | --- | --- |
| Value | −4.3809 | −0.1876 | −0.0872 | 0.3987 | 0.0283 |

Then, the feature value and feeding rate correlation were validated by back-substituting the initial condition data into Equation (11) and comparing the predicted values from the model with the theoretical values. The results are presented in Table 10. These results demonstrate that the theoretical feeding rate and the model-predicted feeding rate are almost identical, with only minimal numerical discrepancies, which indicates that the multiple linear regression model performs exceptionally well on the dataset.

**Table 10.** Comparison of the theoretical and predicted feeding rate.

| Category | $Y_1$ | $Y_2$ | $Y_3$ | $Y_4$ | Theoretical W | Modeling W | Relative Error |
| --- | --- | --- | --- | --- | --- | --- | --- |
| Group 0 | 31.91 | 37.31 | 31.52 | 37.19 | 0 | 0.0034 | / |
| Group 1 | 49.50 | 50.31 | 45.81 | 44.48 | 1.47 | 1.4747 | 0.32% |
| Group 2 | 58.61 | 62.84 | 55.41 | 60.15 | 2.94 | 2.9405 | 0.02% |
| Group 3 | 64.04 | 68.09 | 62.46 | 64.94 | 4.41 | 4.4084 | 0.04% |

### 3.4. Field Test Verification

To validate the accuracy of this feeding rate prediction model, a field test was carried out, and the corresponding field results are shown in Table 11. In Table 11, the theoretical estimated feeding rate for validation group 1 is 0.98 kg/s, while the predicted feed rate calculated by the model is 1.01 kg/s, resulting in a relative error of 3.1%. For validation

group 5, the theoretical value is 5.88 kg/s, and the predicted value is 5.59 kg/s, resulting in a relative error of 4.9%. It is observed that the maximum error occurs in the fifth test, indicating a negative impact of forward velocity on the relative error. With increasing forward velocity, the relative error also increases. The possible reason for this phenomenon could be that as the forward velocity increases, the unevenness of the experimental field has a greater impact on the machine's vibration. Moreover, larger feed rates introduce more uncertainty factors, posing a greater challenge to the stability of the prediction model's performance, and thereby increasing the error in the prediction model. The results of these five validation trials strongly demonstrate that the prediction model established based on time-domain- and harmonic analysis-derived features accurately predicts the feeding rate, with only minimal relative errors compared to the theoretical estimated feeding rate.

**Table 11.** Summary of the field validation experimental data and analysis.

| Verification Group No. | Harvesting Distance (m) | Forward Speed ($m \cdot s^{-1}$) | Predicted Feeding Rate ($kg \cdot s^{-1}$) | Theoretical Feeding Rate ($kg \cdot s^{-1}$) | Relative Error (%) |
|---|---|---|---|---|---|
| Group 1 | 25 | 0.2 | 1.01 | 0.98 | 3.1 |
| Group 2 | 25 | 0.4 | 2.03 | 1.96 | 3.6 |
| Group 3 | 25 | 0.8 | 3.75 | 3.92 | 4.3 |
| Group 4 | 25 | 1.0 | 4.68 | 4.90 | 4.5 |
| Group 5 | 25 | 1.2 | 5.59 | 5.88 | 4.9 |

## 4. Conclusions

The feeding rate is one of the important factors that affect the performance of combine harvesters. By installing vibration acceleration sensors at the bottom of the inclined conveyor, the influence of feeding rate on the vibration characteristics of the combined harvester inclined conveyor was investigated. By analyzing the vibration signals variation under the time domain, it was found that the vibration energy gradually increased as the feeding rate increased, and the peak-to-peak value, standard deviation, and RMS of Sensor 2 were consistently greater than those of Sensor 1, which was attributed to the different installation positions of the sensors and their varying influences from vibration excitation sources. The subsequent analysis of time-domain feature values across experimental groups revealed a gradual increase in peak-to-peak values with feeding rate, albeit with fluctuations, while RMS values increased gradually, reflecting the increase in vibration signal energy. Although peak-to-peak values provide the amplitude range of vibration signals, they are less sensitive to changes in overall shape; in contrast, RMS values more comprehensively reflect the magnitude of a vibration waveform and are more sensitive to a complex waveform.

Furthermore, sampling frequency settings and the number of spectral lines were determined according to the Nyquist theorem in a frequency-domain analysis to ensure accurate signal analysis, while a one-third octave analysis deepened the understanding of the frequency domain characteristics of vibration signals. It was indicated that (1) with an increase in the feeding rate, the average vibration signal energy within the frequency range of 2 Hz to 1000 Hz gradually increased, and the average energy at Sensor 2 was always greater than that at Sensor 1. (2) Sub-bands centered at 315 Hz dominated the frequency domain, exhibiting significance in the one-third octave analysis. The frequency components within this sub-band provided significant vibration intensity, and the amplitude of this sub-band significantly increased at both Sensor 1 and Sensor 2, indicating that vibration intensity within this sub-band increased with an increasing feeding rate. (3) From the changes in total RMS values, it was observed that the total RMS values at both Sensor 1 and Sensor 2 increased with an increasing feeding rate, further demonstrating that increasing the feeding rate led to increased vibration intensity.

Finally, through time-domain and frequency-domain analyses of vibration signals, a model was established considering the time-domain RMS values and total RMS values of

Sensor 1 and Sensor 2, using multiple linear regression to explore the relationship between these features and the feeding rate. Field validation experiments confirmed the model's reliability, with relative errors ranging from 3.1% to 4.9%, indicating that the model can achieve relatively accurate feeding rate prediction results. This research provides a powerful non-invasive diagnostic method for feeding rate control of combine harvesters, offering important theoretical and practical support for improving the efficiency of agricultural machinery and reducing grain losses.

**Author Contributions:** Conceptualization, Z.L.; methodology, Y.Q.; software, Y.Q.; validation, Y.Q.; formal analysis, Y.Q.; investigation, Y.Q.; resources, Z.L.; data curation, Y.Q.; writing—original draft preparation, Y.Q.; writing—review and editing, Z.L.; visualization, Z.S.; supervision, Z.S.; project administration, Z.L.; funding acquisition, Z.L. and Z.S. All authors have read and agreed to the published version of the manuscript.

**Funding:** This research was funded by the National Natural Science Foundation of China (52275251, 51905221) and sponsored by the QingLan Project of Jiangsu Province, China; the Young Talents Cultivation Program of Jiangsu University, China (2022); the Agricultural Science and Technology Support Program of Taizhou, China (TN202208); A project for postdoctoral researchers in Jiangsu Province, China (2019Z106); Priority Academic Program Development of Jiangsu Higher Education Institutions (PADP); and the Key Laboratory of Modern Agricultural Equipment and Technology (Jiangsu University), Ministry of Education, China (MAET202108).

**Institutional Review Board Statement:** Not applicable.

**Informed Consent Statement:** Not applicable.

**Data Availability Statement:** The datasets generated during and/or analyzed during the current study are available from the corresponding author upon reasonable request.

**Conflicts of Interest:** Author Z.L. conducted postdoc research at the company Ruiniu Stock Company Limited, Wuxi, China, under the suppot of A project for postdoctoral researchers in Jiangsu Province, China (2019Z106). The remaining authors declare that the research was conducted in the absence of any commercial or financial relationships that could be construed as a potential conflict of interest.

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
