# Peer review of "Establishment of a Feeding Rate Prediction Model for Combine Harvesters"

_agriculture, doi:10.3390/agriculture14040589_

Round 1

Reviewer 1 Report

Comments and Suggestions for Authors

1.       What is the exact novelty of the work compared to citations from Ref.17-20?

2.       What is noninvasive method? Explain about it.

3.       The authors considered sensors in only two positions. Is it sufficient to study the vibration characteristics?

4.       How time domain signals are converted to frequency? Explain about it.

5.       In feeding rate model what are beta values? Show in a table.

Author Response

What is the exact novelty of the work compared to citations from Ref.17-20?

Answer: Compared to the referenced citations from Ref.17-20, the novelty of this work lies in its focus on developing a non-invasive diagnostic approach to monitor the feed rates of combine harvesters using vibration signals obtained from the inclined conveyor. While citations from Ref.17-20 primarily concentrate on studying the vibration characteristics of combine harvesters and their correlation with factors such as feeding rates or operating conditions, they do not further utilize vibration characteristics to reflect the working status of the combine harvester. In contrast, this work proposes a novel application of vibration signal analysis for real-time monitoring of feeding rates. Therefore, the key innovation of this work lies in its application of vibration signal analysis to monitor the feed rates of combine harvesters, thereby addressing specific needs for optimizing the operational efficiency and reliability of agricultural machinery. We have improved the explanation of this point in the latest manuscript.

What is noninvasive method? Explain about it.

Answer: A non-invasive method refers to a technique or approach that does not require direct contact or alteration of the surface or interior of an object. In scientific and engineering domains, non-invasive methods typically rely on means such as sensors, non-destructive testing techniques, or remote observation to collect data, perform detection, or conduct monitoring without physically intervening or damaging the object. In this study, data collection from the inclined conveyor was accomplished using vibration signals, thereby avoiding any physical intervention or alteration of the combine harvester. This allows for obtaining the necessary information without affecting the normal operation of the machine. 

The authors considered sensors in only two positions. Is it sufficient to study the vibration characteristics?

Answer: Although installing multiple sensors on the inclined conveyor can provide more comprehensive data and enable a more detailed analysis of its vibration characteristics, this poses challenges in data processing. The large volume of data requires more complex data processing and analysis techniques to handle, and the complexity of the feed rate prediction model will also increase exponentially. Furthermore, field validation experiments have shown that the vibration signals collected by two sensors can accurately reflect the feed rate of crops in the inclined conveyor. Therefore, after weighing the pros and cons in practical applications, it was determined to use only two sensors to meet the research requirements. 

How time domain signals are converted to frequency? Explain about it. 

Answer: Converting a time-domain signal into a frequency-domain signal typically involves a mathematical tool called the Fourier transform. The Fourier transform is a method for transforming a signal from the time domain to the frequency domain, allowing the signal to be decomposed into different frequency components. This transformation enables us to analyze the signal in the frequency domain, thereby better understanding the frequency characteristics and components of the signal.

The DH5902 dynamic signal analyzer used in this study not only reliably captures and stores vibration signals but also provides extensive signal data processing capabilities.

In feeding rate model what are beta values? Show in a table. 

Answer: Thank you for your rigorous consideration. We have added a table in the latest manuscript to illustrate this.

Reviewer 2 Report

Comments and Suggestions for Authors

The statements of the Introduction text could be supported by scientific sources (rows 46-62).

Equations 1, 2 and 3 are well known. Or maybe the authors have their own input? If not, it is enough to indicate the literature where to find them. These are well-known things.

All equations must be followed by an explanation of the means of all values. Values in equations 1, 2 and 3 are not explained. When writing explanations, they must be in the same style. The explanation of the values in Equation 6 is in different style.

In Figure 2, the signals for each experimental group are presented as real signals. However, it is very difficult to determine the values of the signals. Please edit to make it clearer.

   The authors state that "the average energy of sensor 2 was always higher than that of sensor 1, indicating differences in signal intensity received by sensors in different positions under the same vibration conditions, which is related to sensor installation positions."

However, the sensors were mounted close to each other. Can the feeding rate of inclined conveyor vary so much?

Maybe "sensor 1 is ... closer to the header and sensor 2 is closer to the threshing device" has more influence? Please explain how the vibrations from the header and threshing device were eliminated from measurements. 

Author Response

The statements of the Introduction text could be supported by scientific sources (rows 46-62).

Answer: We greatly appreciate your valuable suggestions. We have incorporated some new references into the latest manuscript. The references are as follows:

Tama, B.A.; Vania, M.; Lee, S. Recent advances in the application of deep learning for fault diagnosis of rotating machinery using vibration signals. Artif Intell Rev. 2023, 56, 4667–4709.

Mohamad, H.M.; Wan R. Vibration Analysis for Machine Monitoring and Diagnosis: A Systematic Review, Shock and Vibration. 2021, 2021, 25.

Ruiz, C.C.; Jaramillo, V.; Mba, D.; Ottewill, J.; Cao Y. Combination of process and vibration data for improved condition monitoring of industrial systems working under variable operating conditions. Mech Syst Signal Process. 2016, 66, 699–714.

William, T.T; Marie, D.D. Theory of Vibration with Applications, 5rd ed.; tsinghua university press: Bei Jing, Chinese, 2005.

Equations 1, 2 and 3 are well known. Or maybe the authors have their own input? If not, it is enough to indicate the literature where to find them. These are well-known things. All equations must be followed by an explanation of the means of all values. Values in equations 1, 2 and 3 are not explained. When writing explanations, they must be in the same style. The explanation of the values in Equation 6 is in different style.

Answer: Thank you for your nice suggestion. Equations 1, 2, and 3 are well-known. We list these equations in the text along with brief explanations to facilitate better analysis of signal variations in subsequent sections. Regarding the issue of explaining the values in the equations and the different style used, we have addressed this in the latest manuscript.

In Figure 2, the signals for each experimental group are presented as real signals. However, it is very difficult to determine the values of the signals. Please edit to make it clearer.

Answer: We feel sorry for the inconvenience brought to the reviewer; we have re-edited the figures in the latest manuscript to make them clearer.

The authors state that "the average energy of sensor 2 was always higher than that of sensor 1, indicating differences in signal intensity received by sensors in different positions under the same vibration conditions, which is related to sensor installation positions." However, the sensors were mounted close to each other. Can the feeding rate of inclined conveyor vary so much?

Maybe "sensor 1 is ... closer to the header and sensor 2 is closer to the threshing device" has more influence? Please explain how the vibrations from the header and threshing device were eliminated from measurements.  

Answer: Thank you for your attention and inquiry. We fully understand your concerns. Regarding the differences in vibration signal intensity between Sensor 1 and Sensor 2, I would like to further explain the rationale behind installing two sensors and the stability of vibration signals.

Firstly, while a single sensor is capable of reflecting changes in feed rate, we chose to use two sensors to enhance measurement accuracy. Despite these sensors being installed in close proximity to each other, it was the optimal configuration considering the operational environment and structural limitations of the inclined conveyor chute. Sensor 1 is positioned closer to the head, while Sensor 2 is closer to the threshing unit, allowing for a more comprehensive capture of feed rate variations within the conveyor chute. Although the distance between these positions is not significant, simultaneous use of both sensors enables more precise measurement of vibration signals at the bottom of the inclined conveyor chute, thus providing a more accurate reflection of feed rate changes.

Secondly, during field harvesting operations, various components of the combine harvester typically operate at rated speeds, resulting in relatively stable vibration frequencies. Leveraging this stability, we can conduct rational data processing and extract stable feature parameters, such as time-domain root mean square and overall root mean square of harmonics. Additionally, we employed a multiple linear regression model to predict feed rates, comprehensively analyzing the relationship between changes in signal characteristics and feed rates by considering combinations of multiple features, thereby mitigating the effects of vibrations from the head and threshing unit. In the model validation process, we verified the accuracy and reliability of the model using actual data.

Reviewer 3 Report

Comments and Suggestions for Authors

This manuscript is well-structured and containing meaningful research that contributes significantly to the field of agriculture. I noticed a few minor issues that need to be addressed.

On page 4 line 128,the formula sequence number is incorrect. Also, on page 5, there's an issue with the formula on line 166. I suggest the authors double-check the formula number to ensure it matches the rest of the document.

Comments on the Quality of English Language

Minor editing of English language required

Author Response

This manuscript is well-structured and containing meaningful research that contributes significantly to the field of agriculture. I noticed a few minor issues that need to be addressed.

On page 4 line 128,the formula sequence number is incorrect. Also, on page 5, there's an issue with the formula on line 166. I suggest the authors double-check the formula number to ensure it matches the rest of the document.

Answer: Thank you for your recognition and excellent suggestions. We have carefully revised the numbering of the formulas in the latest manuscript.